# SOCS3 Protein Mediates the Therapeutic Efficacy of Mesenchymal Stem Cells against Acute Lung Injury

**DOI:** 10.3390/ijms24098256

**Published:** 2023-05-04

**Authors:** Young Eun Kim, Dong Kyung Sung, Yuna Bang, Se In Sung, Misun Yang, So Yoon Ahn, Yun Sil Chang

**Affiliations:** 1Cell and Gene Therapy Institute, Samsung Medical Center, Seoul 06351, Republic of Korea; duddms920@skku.edu; 2Department of Pediatrics, Samsung Medical Center, Sungkyunkwan University School of Medicine, Seoul 06351, Republic of Korea; dbible@skku.edu (D.K.S.); b.yuna@sbri.co.kr (Y.B.); sein.sung@samsung.com (S.I.S.); misun.yang@samsung.com (M.Y.); soyoon.ahn@samsung.com (S.Y.A.); 3Department of Anatomy and Cell Biology, Sungkyunkwan University School of Medicine, Suwon 16419, Republic of Korea; 4Department of Health Sciences and Technology, SAIHST, Sungkyunkwan University, Seoul 06351, Republic of Korea

**Keywords:** mesenchymal stem cell transplantation, acute lung injury, bacterial infection, suppressor of cytokine signaling proteins

## Abstract

Mesenchymal stem cells (MSCs) have been studied as novel therapeutic agents because of their immunomodulatory properties in inflammatory diseases. The suppressor of cytokine signaling (SOCS) proteins are key regulators of the immune response and macrophage modulation. In the present study, we hypothesized that SOCS in MCSs might mediate macrophage modulation and tested this in a bacteria-induced acute lung injury (ALI) mouse model. The macrophage phenotype was observed in RAW264.7 alveolar macrophages exposed to lipopolysaccharide (LPS) in an in vitro model, and in the ALI mouse model induced by tracheal administration of *Escherichia coli* (1 × 10^7^ CFU in 0.05mL PBS). In LPS-exposed RAW264.7 cells, the levels of markers of M1 macrophages, such as CD86 and pro-inflammatory cytokines (IL-1α, IL-1β, IL-6 and TNF-α), significantly increased, but they significantly reduced after MSC treatment. Meanwhile, the levels of markers of M2 macrophages, such as CD204 and anti-inflammatory cytokines (IL-4 and IL-10), increased after LPS exposure, and further significantly increased after MSC treatment. This regulatory effect of MSCs on M1/M2 macrophage polarization was significantly abolished by SOCS3 inhibition. In the *E*. *coli*-induced ALI model, tissue injury and inflammation in the mouse lung were significantly attenuated by the transplantation of MSCs, but not by SOCS3-inhibited MSCs. The regulatory effect of MSCs on M1/M2 macrophage polarization was observed in the lung injury model but was significantly abolished by SOCS3 inhibition. Taken together, our findings suggest that SOCS3 is an important mediator for macrophage modulation in anti-inflammatory properties of MSCs.

## 1. Introduction

Bacterial pneumonia is a common cause of acute respiratory distress syndrome (ARDS) and acute lung injury (ALI) [1]. Outbursts and uncontrolled inflammatory responses lead to tissue injury and respiratory failure [2]. Despite being a life-threatening condition, there is no cure for serious lung injury.

Macrophages have crucial roles in inflammatory processes, from initiation to resolution [3]. Macrophages usually exist in a resting state, but become activated by activation signals, such as bacterial lipopolysaccharide (LPS) and pro-inflammatory cytokines [4]. This is the classical activated form of macrophages (M1 macrophages). M1 macrophages maintain and promote inflammatory responses to phagocytose bacteria and dead cells, and recruit other inflammatory cells [5]. Macrophages can be alternatively activated (M2 macrophages) by certain cytokines, such as interleukin (IL)-4 and IL-10, to control the immune response and prepare wound healing [5]. Thus, the regulation of macrophage M1–M2 polarization balance would be key to treating acute inflammatory diseases.

Macrophage M1–M2 polarization can be regulated by a suppressor of cytokine signaling (SOCS) via Janus kinase (JAK)/signal transducer and activator of transcription (STAT) pathway [6]. The SOCS family is involved in the negative regulation of cytokine signaling as a regulator of the intensity, duration, and quality of cytokine signaling [7]. SOCS1 and SOCS3 are well-known inhibitors of the JAK/STAT pathway. According to previous studies, SOCS1 showed an anti-inflammatory effect against smoke inhalation and LPS-induced ALI [8,9], and SOCS had an anti-inflammatory role in acute inflammatory arthritis [10]. 

Mesenchymal stem cells (MSCs) are multipotent cells with unique immunoregulatory properties [11,12]. The anti-inflammatory and macrophage modulatory effects of MSCs in various inflammatory diseases have been investigated [13]. However, whether SOCS mediates the therapeutic efficacy of MSCs by modulating macrophage polarization in ALI remains to be elucidated. We hypothesized that MSCs can modulate macrophage M1–M2 polarization via SOCSs, and that this mechanism might be involved in the therapeutic efficacy of MSCs. In this study, we investigated whether the expression levels of SOCSs were upregulated after LPS treatment in MSCs, and whether SOCSs inhibition reduced the macrophage-modulating and anti-inflammatory effects of MSCs in an *Escherichia coli*-induced ALI mouse model.

## 2. Results

### 2.1. Effect of MSCs on Morphometric Changes of LPS-Stimulated Alveolar Macrophages

The morphology of alveolar macrophages (RAW264.7 cells) can vary depending on their phagocytic function and the surrounding environment [14]. In the resting state, RAW264.7 macrophages have a round shape and lack cytoplasmic extensions, while the LPS-stimulated RAW264.7 cells have an enlarged amoeboid shape with numerous fine cytoplasmic extensions (Figure 1A). Quantitative analysis of cell thickness and area showed that the LPS-stimulated RAW264.7 cells were significantly thinner and larger than normal controls. However, these LPS-induced morphological changes were significantly normalized after MSC treatment compared to LPS-stimulated RAW264.7 control cells (Figure 1B).

### 2.2. Effect of MSCs on Regulation of Cytokine Production of LPS-Stimulated Alveolar Macrophages

Pro-inflammatory cytokines, such as IL-1α, IL-1β, tumor necrosis factor (TNF)-α, and IL-6, are secreted by classical macrophages (M1), whilst anti-inflammatory cytokines, such as IL-4 and IL-10, are secreted by regulatory macrophages (M2). In response to LPS stimulation, the levels of IL-1α, IL-1β, TNF-α, and IL-6 in RAW264.7 cells were significantly increased compared to those in normal controls (Figure 1C). However, these increased levels of pro-inflammatory cytokines were significantly reduced after MSC treatment. In the host defense system, the levels of IL-4 and IL-10 were significantly increased in LPS-stimulated RAW264.7 controls compared to those in normal controls, and these increased levels of IL-4 and IL-10 were further significantly increased after MSC treatment compared to LPS-stimulated RAW264.7 controls (Figure 1C).

### 2.3. MSC Transcriptome Analyses after LPS Induction

In the LPS-treated MSCs, 338 genes showed a significant upregulation compared to the non-treated control MSCs. The functions of altered gene expressions were subsequently classified based on gene ontology (Figure 2A) and KEGG pathway (Figure 2B). Gene ontology analysis of the 338 genes revealed that they were related to inflammation response, signal transduction, positive regulation of cell proliferation, cell adhesion, immune response, cytokine-mediated signaling pathway and positive regulation of inflammatory response. KEGG molecular pathways enriched in this group included MAPK, p53, and Toll-like receptor (TLR) signaling pathways. After LPS induction, the significantly upregulated genes in MSCs compared to non-treated control MSCs were related to KEGG pathways, including JAK/STAT signaling pathway, cytokine–cytokine receptor interaction, and TLR signaling pathway (Figure 2B). SOCS1 and SOCS3 genes were commonly involved among the upregulated genes in the biological processes of the negative regulation of inflammatory response, JAK/STAT pathway, and cytokine-mediated signaling identified in gene ontology analysis (Figure 2C). Western blotting analysis confirmed that SOCS1, SOCS2, and SOCS3 expressions were significantly increased in MSCs after LPS induction compared to that in non-treated control MSCs (Figure 2D).

### 2.4. MSC Effect on Alveolar Macrophage Modulation

The levels of pro-inflammatory cytokines IL-1a, IL-1b, IL-6, and TNF-a were significantly elevated in the LPS control group compared to those in the normal control group. The LPS-induced inflammatory-cytokine production was significantly reduced after MSC treatment. However, SOCS1-neutralizing siRNA significantly abolished the effect of MSCs on reducing pro-inflammatory cytokines, and SOCS3-neutralizing siRNA showed a stronger effect than SOCS1-neutralizing siRNA on abolishing the effect of MSCs on reducing pro-inflammatory cytokines (Figure 3A). In contrast, application of SOCS2-neutralizing siRNA had a minimal effect on MSCs. The levels of anti-inflammatory cytokines IL-4 and IL-10 were significantly increased in the LPS control group compared to those in the normal control group, and IL-4 and IL-10 levels were further increased in the MSC- treated LPS group compared to the LPS control group (Figure 3A). SOCS1- and SOCS3-neutralizing siRNA significantly abolished the effect of MSCs on increasing the level of IL-4. SOCS3-neutralizing siRNA showed a stronger effect than SOCS1-neutralizing siRNA on abolishing the effect of MSCs on increasing the level of IL-10. These effects of MSCs were barely affected by SOCS2-neutralizing siRNA.

Furthermore, we observed that the increased level of the M1 specific marker CD86 in the LPS control group was significantly reduced in the MSC-treated LPS group, but the level of CD86 in the MSC-treated group was significantly increased in the SOCS3-suppressed MSC-treated group (Figure 3B). In line with this finding, the level of M2 specific marker CD204 significantly increased in the MSC-treated LPS group compared to the LPS control group, but the level of CD204 in the MSC-treated group was significantly reduced in the SOCS3-inhibited MSC-treated group.

### 2.5. Histological Examination in the E. Coli-Induced ALI Mouse Model

Pathological features of *E*. *coli*-induced ALI lung injury were characterized by alveolar congestion, alveolar wall thickness, alveolar hemorrhage, and neutrophil infiltration, 3 days after the ALI modeling (Figure 4A,B). Injury scores in alveolar congestion, alveolar wall thickness, alveolar hemorrhage, and neutrophil infiltration in *E*. *coli* control group were significantly attenuated after MSC transplantation; however, not after transplantation of SOCS3-suppressed MSCs. Scrambled siRNA treatment did not affect the MSC effect on attenuating lung injury scores in the ALI mouse model.

### 2.6. Macrophage Modulating Function of MSCs

The high levels of the pro-inflammatory cytokines IL-1a, IL-1b, TNF-a, and IL-6 in the *E*. *coli* control group were significantly attenuated after MSC transplantation but not after transplantation of SOCS3-suppressed MSCs (Figure 5A). The expression level of the M1 specific marker CD86 was significantly increased in the *E*. *coli* control group (Figure 5B). After MSC transplantation, the level of CD86 was significantly reduced, but not after transplantation of SOCS3-suppressed MSCs. In line with this result, the level of the M2 specific marker CD163 in the *E*. *coli* control group was not significantly higher than that in the normal control group. However, the level of CD163 was significantly increased after MSC transplantation but not after transplantation of SOCS3-suppressed MSCs.

## 3. Discussion

We previously studied the anti-inflammatory effect of MSCs in murine models of infectious diseases, such as bacterial-induced ALI [15,16] and meningitis [17]. In our previous study [15], we successfully demonstrated that *E. coli*-induced acute lung injury using ICR mouse showed significant increases in lung injury, including alveolar congestion, hemorrhage, neutrophil infiltration, and alveolar wall thickness, as well as inflammatory cytokine levels, which are similar to those observed in clinical ALI/ARDS [18]. The ICR mouse strain is commonly used in in vivo disease models, including acute lung injury, as reported in other previous studies [19,20]. Furthermore, it is known that C57BL/6 mice and ICR mice have similar physiological features and metabolic phenotypes [21]. According to previous studies, the immunomodulatory properties of MSCs are mediated by paracrine factors [16,22]. However, knowledge of the detailed mechanism is required in order for clinical applications of MSC transplantation for the treatment of the inflammatory diseases, including bacterial-induced ALI. In the present study, we demonstrated that LPS-exposed MSCs secreted SOCS1–3, and that MSCs were effective in modulating M1/M2 macrophage polarization via SOCS3. In the bacterial-induced lung injury model, the M1 macrophage ratio greatly increased, while the M2 macrophage ratio slightly or rarely increased depending on the host defense mechanism. However, MSCs modulated the M1/M2 polarization balance by reducing the M1 phenotype and increasing the M2 phenotype of alveolar macrophages. These results indicate that, in bacterial-induced ALI, the anti-inflammatory effect of MSCs might be mediated by modulating M1/M2 macrophage polarization via SOCS3. The role of SOCS3 as a regulator of macrophage polarization and function, as well as the effect of MSCs on modulating macrophages have been investigated [23,24]. However, this is the first study to demonstrate that MSC transplantation is effective in regulating macrophage polarization via SOCS3.

Macrophages are an essential inflammatory cell for host defense during infection. In inflammatory lung injury, pulmonary macrophages can induce and resolve the inflammatory response via an M1/M2 phenotypic shift [25]. After the pro-inflammatory response of M1 macrophages to eliminate pathogens, the role of M2 macrophages in resolution is important for tissue recovery. However, in uncontrolled pulmonary inflammation, the imbalance of M1/M2 macrophages is involved in the pathogenesis of lung injury. According to previous studies, the imbalance in M1/M2 macrophage polarization is closely associated with injury severity and the pathogenesis of inflammatory diseases, such as osteoarthritis [26,27]. This suggests that strategies to shift macrophages into M2 activation can be key to the treatment of inflammatory diseases. In our in vitro and in vivo models of infectious lung injury, we showed that MSCs significantly attenuated pro-inflammatory cytokine levels and alveolar macrophage polarization toward M1 cells, and promoted anti-inflammatory cytokine levels and alveolar macrophage polarization toward M2 cells. Additionally, in our lung injury model, we observed that SOCS3 expression was upregulated in LPS-exposed MSCs, and that SOCS3-suppressed MSCs were ineffective in modulating macrophage polarization and anti-inflammation. Furthermore, we observed that SOCS1 and SOCS2 were significantly upregulated in LPS-exposed MSCs, but the anti-inflammatory effect of MSCs was rarely affected by the suppression of SOCS1 and SOCS2. This suggests that although MSCs expressed SOCS1–3, among the SOCS family members, SOCS3 might be the dominant mediator of the anti-inflammatory effect in MSCs.

MSCs exert their role through paracrine activity. Thus, MSCs preconditioned in conditions similar to those of the disease microenvironment can play roles, such as in immunomodulation, more efficiently when transplanted into disease models [28]. We have previously demonstrated that preconditioned MSCs, compared to naïve MSCs, produce higher secretome levels with richer cargo contents associated with disease treatment [29], and show a better therapeutic efficacy [30]. In the present study, we used LPS-preconditioned MSCs to maximize the paracrine efficacy of MSCs in the bacterial-induced lung injury model. We observed that MSCs preconditioned with LPS significantly attenuated inflammation and improved tissue injury. Further studies are needed to compare different pretreatment methods to enhance the immunomodulatory function of MSCs and thus determine the most effective method of MSC transplantation in bacterial-induced lung injury models.

In summary, our study showed that MSCs are effective in modulating M1/M2 macrophages functionally as well as phenotypically and in attenuating inflammation in a bacterial lung infection model. By analyzing the changing patterns of genes associated with immunomodulation, we observed that the expression levels of the SOCS family were upregulated in MSCs after LPS exposure. Next, SOCS3 was investigated as an important mediator in macrophage regulation in MSCs. These findings suggest that, by improving the valance of M1/M2 macrophages, MSC transplantation could be a feasible treatment for inflammatory diseases, such as infectious lung injury. It would be necessary to perform cytokine profiling in SOCS3-overexpressing MSCs to demonstrate the effectiveness of SOCS3 and to develop therapeutically enhanced MSCs overexpressing SOCS3. We plan to investigate whether the anti-inflammatory effect of MSCs can be enhanced after SOCS3 overexpression in this infectious lung injury model. Based on the results of this study, in a further study, we will develop SOCS3-overexpressing MSCs applicable to clinical acute lung injury and evaluate their therapeutic efficacy.

## 4. Materials and Methods

### 4.1. Preparation of MSCs

This study was approved by the Institutional Review Board of the Samsung Medical Center. Human umbilical cord blood (hUCB) were collected after obtaining informed consent from normal full-term pregnant mothers. hUCB-derived MSCs were isolated from a single donor, expanded and cultured as previously described [31,32]. Briefly, MSCs were cultured in MEMα (Gibco, Grand Island, NY, USA) with 10% fetal bovine serum (FBS; Gibco) in a humidified incubator with 5% CO_2_ at 37 °C, and the culture media was changed twice a week. During cell culture, we used the same product lot numbers of cell culture materials such as FBS and cell culture flask. MSC characterization and differentiation potentials were confirmed using previously established protocols [31]. MSCs were positive for MSC cell surface markers (CD73 and CD105) but negative for hematopoietic markers (CD14, CD34 and CD45) in flow cytometric analyses, and MSCs can differentiate into osteoblast, chondrocyte, and adipocytes in in vitro differentiation assays, as previously described [31]. MSCs at passage 5–6 were used in the study. Before transplantation, MSCs reaching approximately 90% confluency were preconditioned with LPS (1 µg/mL) for 6 h.

### 4.2. SOCSs Suppression in MSCs

Small interfering (siRNA)-mediated knockdown was used to suppress SOCS expression in MSCs. SOCS1-, 2-, and 3- and scrambled-siRNAs were purchased from Santa Cruz Biotechnology (Santa Cruz, California, USA). MSCs were transfected with siRNA targeting SOCS 1–3 using Lipofectamine (Invitrogen, Carlsbad, CA, USA). As a negative control, scrambled siRNA was transfected into MSCs using the same method. The SOCS3 suppression was confirmed using western blotting (Appendix A).

### 4.3. In Vitro Models of LPS-Induced Inflammation in Alveolar Macrophages

Alveolar macrophages (RAW264.7 cells; Korean Cell Line Bank, Seoul, Republic of Korea) were cultured (1 × 10^5^ cells) in RPMI 1640 medium (Gibco BRL, Grand Island, NY, USA) containing 10% FBS with penicillin–streptomycin (100 U/mL Gibco BRL). At approximately 90% confluency, the RAW264.7 cells were stimulated with LPS (1 µg/mL) for 24 h in serum-free media. In the MSC-treated group, MSCs were co-cultured at a 10:1 ratio with LPS induction.

### 4.4. Quantification of Morphometric Changes in Alveolar Macrophages

Alveolar macrophages were seeded in six-well plates at a density of 5 × 10^4^ cells/well. Alveolar macrophages were stimulated with LPS (200 ng/mL) in serum-free media. In the MSC-treated group, MSCs were co-cultured with LPS-induced alveolar macrophages and imaged using the HoloMonitor cell imaging system (Holographic Imaging, Lund, Sweden) with a microscope (LX2-KSP; Olympus, Shinjuku, Tokyo, Japan), with images captured by a CCD camera. The M4 Studio software 2.6.2 was used to analyze the data.

### 4.5. Microarray Analysis

Changes in gene expression in MSCs after LPS induction were analyzed using bead-based microarray analyses (Agilent, Palo Alto, CA, USA), as previously described [15]. Briefly, total RNA of MSCs were isolated using Trizol kit (Invitrogen) and the RNA was hybridized to Agilent Human Oligo Micaroarray (60 K) chips. The hybridized images were scanned using Agilent DNA microarray scanner and quantified with Feature Extraction Software version (10.7) (Agilent). Normalization of all data and selection of fold-changed genes was performed using GeneSpringGX 7.3 (Agilent). A gene expression change of 3-fold or more and a *p*-value of 0.05 or less were considered significant changes in gene expression. The list of relevant genes in the DAVID online database (http://david.abcc.ncifcrf.gov/ (accessed on 17 February 2023)), Medline (http://www.ncbi.nlm.nih.gov/ (accessed on 17 February 2023)), and the KEGG database (http://www.genome.jp/kegg/ (accessed on 17 February 2023)) were used to analyze functional annotations.

### 4.6. Preparation of Bacteria

*E*. *coli* was used for infection to establish an in vivo model of bacterial-induced ALI, because Gram-negative bacteria are a common cause of lung infection [33]. We used the *E*. *coli* strain E69 (a gift from Dr. Kwang Sik Kim, Johns Hopkins Hospital, Baltimore, MD, USA), as described in our previous study [15,34]. Briefly, *E*. *coli* was cultured in brain heart infusion broth (BHI; Difco Laboratories, Detroit, MI, USA) at 37 °C overnight. The bacteria were then sub-cultured for 1 h to the mid-log growth phase. The broth media containing *E*. *coli* was washed twice (centrifugation for 10 min at 5000× *g*) in phosphate-buffered saline (PBS). After the optical density was measured, the bacterial sample was adjusted to a final concentration of 10^7^ colony-forming units (CFU) in 0.05 mL PBS. The bacterial concentration was reconfirmed by counting the number of single colonies cultured by spread plating.

### 4.7. In Vivo Model of Bacterial-Induced ALI

For the in vivo experiments, eight-week-old male ICR mice (Orient Bio, Inc., Seoul, Republic of Korea) were anesthetized with a mixture of ketamine and xylazine (45 mg/kg and 8 mg/kg, respectively) by intraperitoneal (i.p.) injection. As previously described [15], the mice were endotracheally intubated to inject the *E*. *coli* into the lungs. Briefly, each mouse was placed in an inclined position at a 70° angle, and their vocal cords were visualized using an otoscope. *E*. *coli* at 10^7^ CFU in 0.05 mL PBS was injected, and the bacterial distribution was evened out by a 2 cm H_2_O-pressure air inflation. The *E*. *coli* injection procedure was completed within 30 s for each mouse. For MSC transplantation, MSCs (1 × 10^5^ cells in 0.05 cc PBS) were endotracheally transplanted within two hours after *E.coli* injection. Mice in the control groups received an equal volume of PBS. Antibiotics (ceftriaxone; 100 mg/kg, once daily) were administered i.p. consecutively for 3 days after bacterial-induced lung injury. 

### 4.8. Tissue Preparation

Lung tissues and bronchoalveolar lavage (BAL) fluid were obtained at 3 d post-injury after the mice were sacrificed under deep pentobarbital anesthesia (60 mg/kg, i.p.). As previously described [15], the trachea was carefully exposed, and a catheter was inserted into the trachea to obtain BAL fluid. The depth of insertion of the catheter was monitored to avoid damaging the lung structures. For histological analyses, lung tissues were extracted and inflated with normal saline at a constant pressure of 20 cm H_2_O. The inflated lungs were immersed in 4% paraformaldehyde for fixation. The lung tissues were paraffin-embedded and sectioned at 4 µm for histological observation of the tissue injury.

### 4.9. Lung Injury Scores

The paraffin-sectioned lung tissues were stained with hematoxylin and eosin. Lung injury was scored as described previously [15]. Briefly, two sections per mouse were randomly selected, and three random microscopic fields of the lung were evaluated by a blinded observer. Lung injury was scored according to the findings in the following four categories: alveolar congestion, hemorrhage, neutrophil infiltration into the airspace or vessel wall, and alveolar wall thickness/hyaline membrane formation. Each category was graded on a five-point scale: 0 = minimal injury; 1 = injury in up to 25% of the field; 2 = injury in up to 50% of the field; 3 = injury in up to 75% of the field; and 4 = diffused injury [35].

### 4.10. Western Blots

The levels of SOCS 1–3 were measured using Western blotting. Membranes were blocked and incubated with polyclonal antibodies against SOCS1, SOCS2, and SOCS3 (Cell Signaling Technology, Inc., Beverly, MA, USA). The membranes were washed with PBS containing 0.5% Tween-20 (PBS-T) and then incubated with the secondary antibodies, anti-rabbit horseradish peroxidase-conjugated immunoglobulin G (1:2000), for 1 h at room temperature with agitation. Following washing with PBS-T, the protein bands were detected using the ECL Select chemiluminescence reagent (GE Healthcare Life Sciences, Piscataway, NJ, USA), and images were acquired using X-ray film.

### 4.11. Enzyme-Linked Immunosorbent Assay

The levels of pro-inflammatory cytokines IL-1α, IL-1β, IL-6, and TNF-α and anti-inflammatory cytokines IL-4 and IL-10 were measured in conditioned media of cultured cells and in BAL fluid of mouse lungs. The cytokine levels were measured using commercial enzyme-linked immunosorbent assay kits (R&D Systems, Minneapolis, MN, USA).

### 4.12. Fluorescence Activated Cell Sorting (FACS)

For FACS analysis of RAW264.7 cells, anti-CD86 and anti-CD204 (BioLegend, San Diego, CA, USA) were used as FACS analysis antibodies for macrophage M1/M2 specification. For FACS analysis of the mouse BAL fluid, BAL fluid was filtered through a 40 µm cell strainer and centrifuged for 10 min at 450× *g* at 4 °C. The cell pellet was re-suspended in RBC lysis buffer (Sigma, USA) and centrifuged (450× *g* at 4 °C). The cell pellet was re-suspended in PBS and again under the same conditions. Anti-CD86 and CD163 (BioLegend) were used as FACS analysis antibodies to stain the BAL fluid. Both CD204 and CD163 are commonly used as markers for M2 macrophages [36]. CD204-positive and CD163-positive cells exhibited similar trends in immunostaining and cell counting in lung tissues; we used CD163 as the marker for M2 macrophage specification in mouse BAL samples [36].

### 4.13. Statistical Analyses

Data are presented as the mean ± standard error of the mean (SEM). Statistical comparisons between groups were evaluated using one-way analysis of variance (ANOVA) and Tukey’s post hoc analysis. All data were analyzed using the SAS software (version 9.4; SAS Institute, Cary, NC, USA). *p*-values less than 0.05 were considered statistically significant.

## Figures and Tables

**Figure 1 ijms-24-08256-f001:**
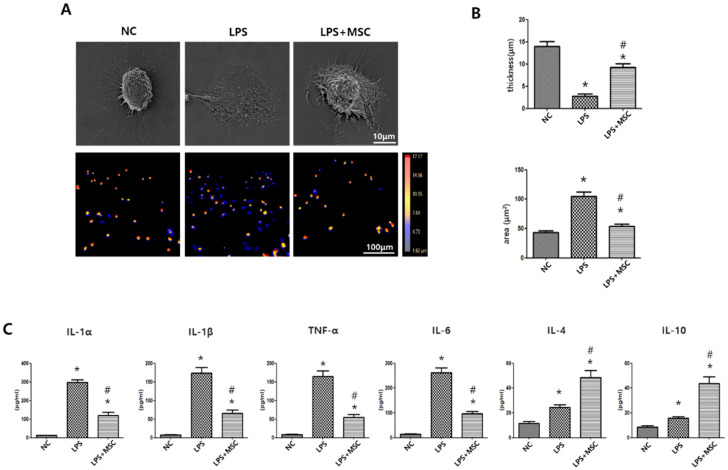
Phenotypic changes of LPS-stimulated alveolar macrophages after MSC treatment. (**A**) Representative images of RAW264.7 alveolar macrophages, taken at 2000× magnification using scanning electron microscopy (upper panel; bar 10 µm) and taken at 100× magnification using HoloMonitor cell imaging system (bottom panel; bar 100 µm). The color bar indicates cell height (thickness); higher is red, lower is blue. (**B**) Quantification of thickness and area of the cells. (**C**) Levels of pro-inflammatory cytokines IL-1α, IL1-β, TNF-α, and IL-6, and levels of anti-inflammatory cytokines IL-4 and IL10. Data are given as mean ± SEM. *, *p* < 0.05 vs. NC. #, *p* < 0.05 vs. LPS. NC, normal control; LPS, LPS-stimulated control; LPS + MSC, LPS stimulation with MSC treatment (*n* = 5 per each analyses).

**Figure 2 ijms-24-08256-f002:**
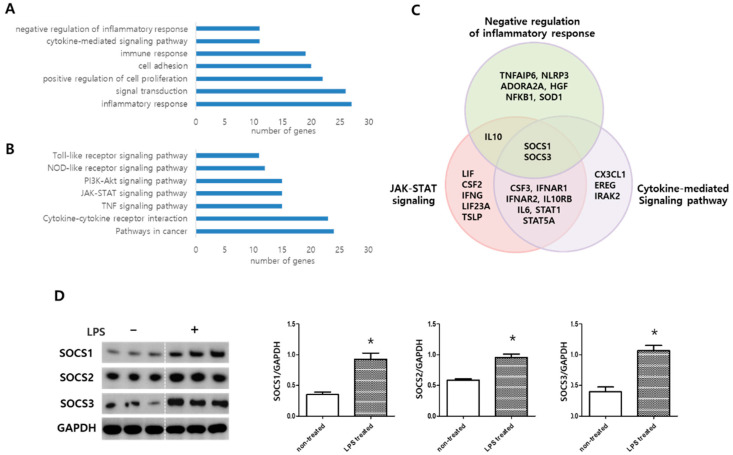
MSC transcriptome analyses. (**A**) Enriched functional categories and the number of genes (*x*-axis) in each category among the 338 significantly upregulated genes in MSCs after LPS (1 µg/mL) induction for 6 h compared to non-treated control MSCs. (**B**) KEGG molecular pathway analysis and the number of genes (*x*-axis) in each category among the 338 significantly upregulated in MSCs after LPS induction compared to non-treated control MSCs. (**C**) Venn diagram analyses indicating significantly upregulated genes in MSCs in the microarray analyses. (**D**) Validation of upregulations of SOCS1–3 expressions by Western blotting. The same lysates were loaded into multiple wells and developed separately for analyses. Levels of SOCS1–3 normalized to GAPDH loading control (Full-length Western blots are shown in Appendix A). Data are given as mean ± SEM. *, *p* < 0.05 vs. NC. NC, non-treated control MSCs; LPS, LPS-induced MSCs (*n* = 3 per each analyses).

**Figure 3 ijms-24-08256-f003:**
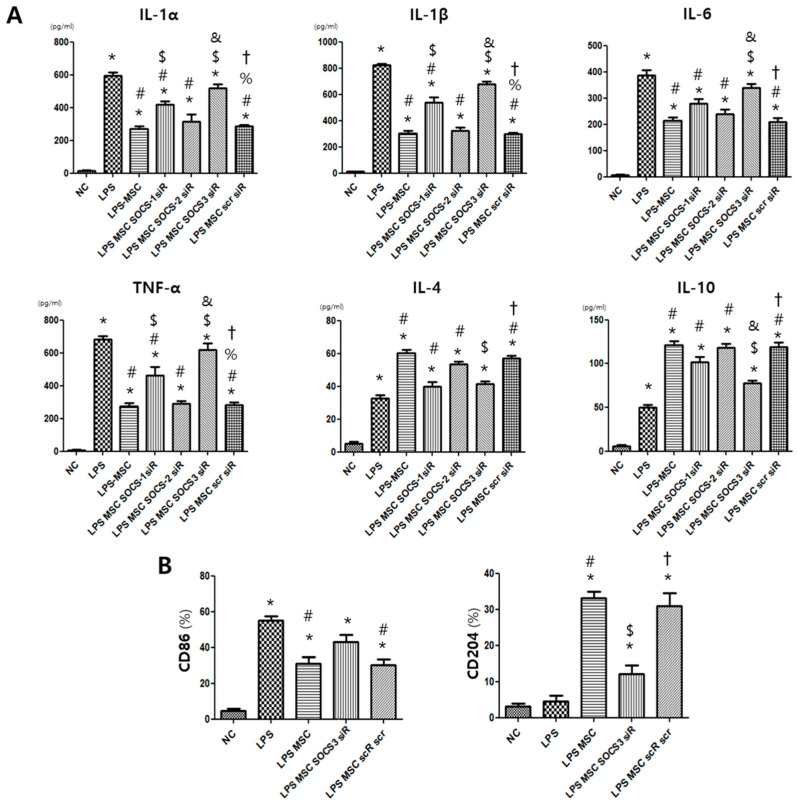
Levels of inflammatory cytokines and cell surface markers specific to M1 and M2 alveolar macrophages. (**A**) Levels of M1-associated pro-inflammatory cytokines IL-1α, IL-1β, IL-6, and TNF-α and levels of M2-associated anti-inflammatory cytokines IL-4 and IL-10 in RAW264.7 cells. (**B**) Percentages of markers for M1 and M2 alveolar macrophages (CD86 and CD204, respectively) in RAW264.7 cells. Data are given as mean ± SEM. *, *p* < 0.05 vs. NC. #, *p* < 0.05 vs. LPS. $, *p* < 0.05 vs. LPS + MSC. %, *p* < 0.05 vs. LPS + MSC SOCS1 siR. &, *p* < 0.05 vs. LPS + MSC SOCS2 siR. †, *p* < 0.05 vs. LPS + MSC SOCS3 siR. NC, normal control; LPS, LPS-stimulated control; LPS + MSC, LPS stimulation with MSC treatment.; LPS + MSC SOCS1 siR, LPS stimulation with SOCS1 siRNA-transfected MSCs; LPS + MSC SOCS2 siR, LPS stimulation with SOCS2 siRNA-transfected MSCs; LPS + MSC SOCS3 siR, LPS stimulation with SOCS3 siRNA-transfected MSCs; LPS + MSC scR scr, LPS stimulation with scrambled siRNA-transfected MSCs (*n* = 5 per each analyses).

**Figure 4 ijms-24-08256-f004:**
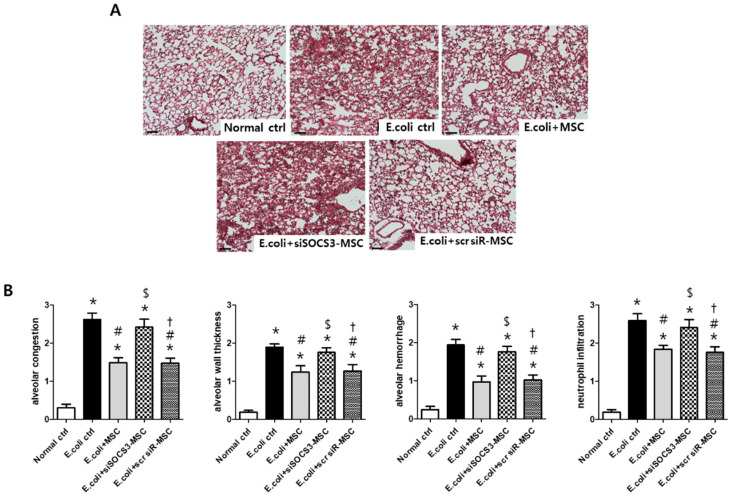
Gross histological evaluation. (**A**) Representative optical microscopy images of mouse lung sections stained with hematoxylin and eosin, taken at 200× magnification (bar, 25 µm). (**B**) Lung injury scores measured in four categories; alveolar congestion, alveolar wall thickness, alveolar hemorrhage, and neutrophil infiltration. Data are given as mean ± SEM. *, *p* < 0.05 vs. Normal ctrl. #, *p* < 0.05 vs. *E*. *coli* ctrl. $, *p* < 0.05 vs. *E*. *coli* + MSC, †, *p* < 0.05 vs. *E*. *coli* + MSC SOCS3 siR. Normal ctrl, normal control; *E*. *coli* ctrl, *E*. *coli*-induced ALI control; *E*. *coli* + MSC, *E*. *coli*-induced ALI with MSC transplantation.; *E*. *coli* + MSC SOCS3 siR, *E*. *coli*-induced ALI with SOCS3 siRNA-transfected MSCs; *E*. *coli* + MSC scR scr, *E*. *coli*-induced ALI with scrambled siRNA-transfected MSCs (*n* = 6 per each analyses).

**Figure 5 ijms-24-08256-f005:**
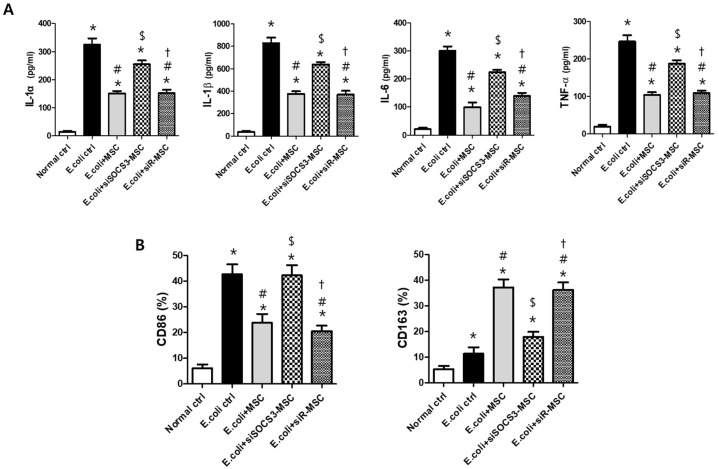
Lung inflammation and modulation of M1/M2 phenotype in alveolar macrophages measured in BAL fluid 3 d after *E*. *coli*-induced ALI. (**A**) Levels of pro-inflammatory cytokines IL-1α, IL-1β, IL-6, and TNF-α. (**B**) Percentages of markers for M1 and M2 alveolar macrophages (CD86 and CD163, respectively) in BAL fluid. Data are given as mean ± SEM. *, *p* < 0.05 vs. Normal ctrl. #, *p* < 0.05 vs. *E*. *coli* ctrl. $, *p* < 0.05 vs. *E*. *coli* + MSC, †, *p* < 0.05 vs. *E*. *coli* + MSC SOCS3 siR. Normal ctrl, normal control; *E*. *coli* ctrl, *E*. *coli*-induced ALI control; *E*. *coli* + MSC, *E*. *coli*-induced ALI with MSC transplantation.; *E*. *coli* + MSC SOCS3 siR, *E*. *coli*-induced ALI with SOCS3 siRNA-transfected MSCs; *E*. *coli* + MSC scR scr, *E*. *coli*-induced ALI with scrambled siRNA-transfected MSCs (*n* = 6 per each analyses).

## Data Availability

The data generated in this study are included in the paper.

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
