# Peer review of "SOCS3 Protein Mediates the Therapeutic Efficacy of Mesenchymal Stem Cells against Acute Lung Injury"

_ijms, 2023, doi:10.3390/ijms24098256_

Round 1

Reviewer 1 Report

The article entitled “SOCS3 Protein Mediates the Therapeutic Efficacy of Mesenchymal Stem Cells Against Acute Lung Injury”, studies the role of SOCS, especially SOCS3 in macrophage modulation. They have used a bacteria-induced acute lung injury (ALI) mouse model as well as alveolar macrophages (RAW264.7) exposed to lipopolysaccharide (LPS) model to study this. In LPS-exposed RAW264.7 cells, the levels of M1 macrophage markers such as CD86 and pro-inflammatory cytokines (IL-1α, IL-1β, IL-6 23 and TNF-α) increased but significantly reduced after MSC treatment. Meanwhile, the levels of M2 macrophage markers such as CD204 and anti-inflammatory cytokines (IL-4 and 25 IL-10) increased after LPS exposure, and further after MSC treatment. This regulatory effect of MSCs on M1/M2 macrophage polarization was significantly abolished by SOCS3 inhibition. The authors also found in the E. coli-induced ALI model, that the tissue injury and inflammation in the mouse lung were significantly attenuated by transplantation of MSCs but not by SOCS3-inhibited MSCs and the same with the regulatory effect of MSCs on M1/M2 macrophage polarization.

I have included my comments below and would recommend major revision before this paper is accepted for publication.

Comments:

MSC culture and characterization data (morphology, FACS/immunocytochemistry for MSC markers & adipo /osteo/chondro specific staining) are required.

BAL macrophages- do you see the same morphological changes that were observed in RAW cells in the BAL macrophages? SEM data?

Please provide M1/M2 staining in lung tissue to see differences in M1/ M2 expression and M1/M2 polarization.

Translational potential of this work. Data with SOCS3 inhibitor? Otherwise include it in the discussion.

Lines 49-50 – Insert references.

Line 59 – ‘The anti-inflammatory and macrophage modulatory effects of MSCs in various inflammatory diseases have been investigated’ - Include/ insert references.

Lines 106-107 -The gene expressions of SOCS 1 and 3 were significantly upregulated in MSCs after LPS induction…..Is this explanation for microarray? If so, clarify in the text. If not correct it. When we read it, it feels like an explanation for PCR data.

Line 292 – correct ‘estavlished an..

Line 304 – why specifically ICR mouse was used? Any reason? Why not C57Bal?

Figure 1: Phenotypic changes – what do the lower 3 images (colored/stained) in the ’A ‘ panel represent? What do the colors as well as the color bar indicate? Why is it important in this context? What additional information do you get from there?

Figure 2- Panel A - what is the X axis?

Figure 2 legend – Correct SOC1-3 to ‘SOCS 1-3’ in two places.

How many experimental replicates were performed? Include n= how many? under each figure legend.

Line 187 – Figure legend – I think ‘E. coli + MSC SOCS1 siR’ needs to be corrected to ‘E. coli + MSC SOCS3 siR’. Don’t see any ‘E. coli + MSC SOCS1’ in any figure labels.

Also, did you look at ‘E. coli + MSC SOCS1’ and ‘E. coli + MSC SOCS1’ BAL cytokine profile?

Why CD204 was used for in vitro studies and CD163 for BAL M2 macrophages. Was it because the same marker behaved differently for the BAL and in vitro experiments?

Did you get a chance to look at the effect of SOCS3 on cellular apoptosis?

Overexpression of SOCS3 to see inflammatory cytokine profile can be looked at.

Good except in a few places

Author Response

  • REVIEWER’S COMMENTS & SUGGESTIONS

  • # Reviewer 1

The reviewer commented, “The article entitled “SOCS3 Protein Mediates the Therapeutic Efficacy of Mesenchymal Stem Cells Against Acute Lung Injury”, studies the role of SOCS, especially SOCS3 in macrophage modulation. They have used a bacteria-induced acute lung injury (ALI) mouse model as well as alveolar macrophages (RAW264.7) exposed to lipopolysaccharide (LPS) model to study this. In LPS-exposed RAW264.7 cells, the levels of M1 macrophage markers such as CD86 and pro-inflammatory cytokines (IL-1α, IL-1β, IL-6 23 and TNF-α) increased but significantly reduced after MSC treatment. Meanwhile, the levels of M2 macrophage markers such as CD204 and anti-inflammatory cytokines (IL-4 and 25 IL-10) increased after LPS exposure, and further after MSC treatment. This regulatory effect of MSCs on M1/M2 macrophage polarization was significantly abolished by SOCS3 inhibition. The authors also found in the E. coli-induced ALI model, that the tissue injury and inflammation in the mouse lung were significantly attenuated by transplantation of MSCs but not by SOCS3-inhibited MSCs and the same with the regulatory effect of MSCs on M1/M2 macrophage polarization.”

1-1) The reviewer stated, “MSC culture and characterization data (morphology, FACS/immunocytochemistry for MSC markers & adipo /osteo/chondro specific staining) are required.”

-> As the reviewer suggested, in the revised manuscript, we have provided detailed information on the MSCs used, including confirmation of their characterization and differentiation potential using established protocols (Cytotherapy, 2015; 17: 1025-1035). Flow cytometric analysis showed that the MSCs were positive for MSC cell surface markers (CD73 and CD105), but negative for hematopoietic markers (CD14, CD34 and CD45). Additionally, the MSCs were capable of differentiating into osteoblasts, chondrocytes and adipocytes in in vitro differentiation assays, as previously described (Cytotherapy, 2015; 17: 1025-1035). These details have been incorporated into the Materials and Methods section of the revised manuscript. (line 300-305)

1-2) The reviewer stated, “BAL macrophages- do you see the same morphological changes that were observed in RAW cells in the BAL macrophages? SEM data?”

-> In our animal study, we did not observe the morphological changes of BAL macrophages. The reviewer suggested that images showing the morphological changes in BAL macrophages would support our study. However, due to limited resources such as experimental animals, we primarily focused on testing the macrophage modulating effect of MSCs via SOCS3 in RAW264.7 cells in an in vitro study. Finally, we confirmed the attenuation of lung inflammation by MSCs depending on SOCS3 expression using ELISA and FACS methods in bacterial acute lung injury model.

1-3) The reviewer stated, “Please provide M1/M2 staining in lung tissue to see differences in M1/ M2 expression and M1/M2 polarization.”

->  We agree with the reviewer’s opinion that directly staining and analyzing M1/M2 in the histologic examination is the definitive way to determine M1/M2 expression and polarization. However, due to the limited resource, instead of histologic evaluation of lung tissue of in vivo animal experiments with each marker of M1/M2 alveolar macrophages, we conducted an analysis of the M1/M2 portion in BAL fluid using FACS (as Figure 3B), which is generally considered as an effective method for both cell-type specifications and for reflecting the state of the lungs in in vivo animal experiments.

1-4) The reviewer stated, “Translational potential of this work. Data with SOCS3 inhibitor? Otherwise include it in the discussion.”

-> Since there is currently no SOCS3-specific inhibitor available, we did not use a specific SOCS3 inhibitor in the present study.  However, based on the results obtained, we are planning to develop SOCS-overexpressing MSCs that are applicable in clinical cases of acute lung injury. In further studies, we will evaluate the therapeutic efficacy of these cells. We have incorporated this into the Discussion section in the revised manuscript. (line 288-290)

1-5) The reviewer stated, “Lines 49-50 – Insert references.”

-> We have inserted the reference (Front Immunol. 2014 Nov 28;5:614) [New Reference #6] (line 51-53).

1-6) The reviewer stated, “Line 59 – ‘The anti-inflammatory and macrophage modulatory effects of MSCs in various inflammatory diseases have been investigated’ - Include/ insert references.”

-> We have inserted the reference (J Immunotoxicol. 2020 Dec;17(1):21-30) [New Reference #13] (line 60-61).

1-7) The reviewer stated, “Lines 106-107 -The gene expressions of SOCS 1 and 3 were significantly upregulated in MSCs after LPS induction…..Is this explanation for microarray? If so, clarify in the text. If not correct it. When we read it, it feels like an explanation for PCR data.”

-> We appreciate the reviewer’s comments. We have clarified the sentences describing Figure2 in the Result section in the revised manuscript. (line 109-124)

1-8) The reviewer stated, “Line 292 – correct ‘estavlished an..’”

-> We have corrected the typo ‘estavlished an..’ to ‘established an..’. (line 341).

1-9) The reviewer stated, “Line 304 – why specifically ICR mouse was used? Any reason? Why not C57Bal?”

-> In our previous study (Respir Res. 2011 Aug 15;12(1):108) we successfully demonstrated that E.coli-induced acute lung injury using ICR mouse showed significant increases in lung injury including alveolar congestion, hemorrhage, neutrophil infiltration, and alveolar wall thickness, as well as inflammatory cytokine levels, which are similar to those observed in clinical ALI/ARDS (Arch Pathol Lab Med. 2010 May;134(5):719-27). The ICR mouse strain is commonly used in in vivo disease models, including acute lung injury, as reported in other previous studies (Nutrients. 2020 Jun; 12(6): 1742 and Inflamm Res. 2022 Jun;71(5-6):627-639). Furthermore, it is known that C57BL/6 mice and ICR mice have similar physiological features and metabolic phenotypes (Lab Anim Res 2017: 33(2), 140-149). We have incorporated this into the Discussion section in the revised manuscript. (line 220-228)

1-10) The reviewer stated, “Figure 1: Phenotypic changes – what do the lower 3 images (colored/stained) in the ’A ‘ panel represent? What do the colors as well as the color bar indicate? Why is it important in this context? What additional information do you get from there?”

-> As the reviewer commented, we have added more explanation to the figure legend of Figure 1A. The color bar in Figure 1A represents the cell height, with red indicating higher and blue indicating lower values. Specifically, a red cell indicates a round shape, which represents the inactivated form of alveolar macrophages, while a blue cell represents a more flattened shape, which represents the activated form of alveolar macrophages. We have incorporated this into the figure legend in the revised manuscript. (line 88-89)

1-11) The reviewer stated, “Figure 2- Panel A - what is the X axis?”

-> In the revised manuscript, Figure 2A and B shows the results of the enriched functional categories and the KEGG molecular pathway analysis, respectively. X-axis represents the number of genes in each pathway among the 338 significantly upregulated in LPS-induced MSCs compared to non-treated control MSCs. We have incorporated this into Figure 2 and the figure legend in the revised manuscript. (line 128)

1-12) The reviewer stated, “Figure 2 legend – Correct SOC1-3 to ‘SOCS 1-3’ in two places.”

-> We appreciate the reviewer's detailed comments on our typos. We have corrected ‘SOC1-3’ to ‘SOCS 1-3’ in Figure 2 legend of the revised manuscript. (line 134 and 135)

1-13) The reviewer stated, “How many experimental replicates were performed? Include n= how many? under each figure legend.”

-> Each data represents the average of duplicate measurements. For in vitro experiments, we used 5 wells per group, and for in vivo experiments, we used 6 mice per group. We have incorporated this information into each figure legend in revised manuscript.

1-14) The reviewer stated, “Line 187 – Figure legend – I think ‘E. coli + MSC SOCS1 siR’ needs to be corrected to ‘E. coli + MSC SOCS3 siR’. Don’t see any ‘E. coli + MSC SOCS1’ in any figure labels.”

-> We appreciate the reviewer's detailed comment. We have corrected ‘E. coli + MSC SOCS1 siR’ to ‘E. coli + MSC SOCS3 siR’. (line 192and 214)

1-15) The reviewer stated, “Also, did you look at ‘E. coli + MSC SOCS1’ and ‘E. coli + MSC SOCS1’ BAL cytokine profile?”

-> We did not analyze the BAL cytokine profile of the ‘E. coli + MSC SOCS1 siR’ group. Instead, we only analyzed BAL cytokine levels in the ‘E.coli + MSC SOCS3 siR’ group and its controls since SOCS3 was found to be the primary mediator among SOCS1‒3 in the immunomodulatory effect of MSCs in this study.

1-16) The reviewer stated, “Why CD204 was used for in vitro studies and CD163 for BAL M2 macrophages. Was it because the same marker behaved differently for the BAL and in vitro experiments?”

-> Both CD204 and CD163 are commonly used as markers for M2 macrophages (PLoS One. 2014 Jan 30;9(1):e87400). We conducted initially in vitro studies using CD204. However, when analyzing the mouse BAL samples from in vivo experiments, we did not have anti-CD204 antibodies available for FACS analysis. Therefore, based on a previous study (PLoS One. 2014 Jan 30;9(1):e87400), which found that CD204-positive and CD163-positive cells exhibited similar trends in immunostaining and cell counting in lung tissues, we used CD163 as the marker for M2 macrophage specification in mouse BAL samples. We incorporated this into the Materials and Methods section in our revised manuscript. (line 407-411)

1-17) The reviewer stated, “Did you get a chance to look at the effect of SOCS3 on cellular apoptosis?”

-> We appreciate the reviewer’s insightful comment. We did not investigate the effect of SOCS3 on cellular apoptosis in the present study. Previous research has shown that SOCS3 plays a role in immune regulation by interacting with Jak/STAT signaling pathway (Hum Immunol. 2015 Oct;76(10):775-80) and inhibit apoptosis by negatively regulating the NFκB pathway (Cell Death Dis. 2012 Jun 28;3(6):e334). For example, a research group demonstrated that SOCS3 protein significantly reduced apoptosis as well as inflammation in acute liver injury (Nat Med. 2005 Aug;11(8):892-8). However, in our study, we focused on the macrophage modulation effect of SOCS in MSCs to investigate the key mediator involved in their anti-inflammatory mechanism.

1-18) The reviewer stated, “Overexpression of SOCS3 to see inflammatory cytokine profile can be looked at.”

-> As the reviewer suggested, it would be necessary to perform cytokine profiling in SOCS3-overexpressing MSCs to demonstrate the effectiveness of SOCS3 and to develop therapeutically enhanced MSCs overexpressing SOCS3. In our further study, we plan to investigate whether the anti-inflammatory effect of MSCs can be enhanced after SOCS3 overexpression in this infectious lung injury model. We have incorporated this into the discussion section of the revised manuscript. (line 284-288)

Reviewer 2 Report

Kim et al in this study wanted to study the involvement of SOCS3 protein in  the therapeutic efficacy of  Mesenchymal Stem Cells against Acute Lung Injury. 

In my opinion this paper is interesting and well done but lacks of some point that could be explain better. 

1. MSC's source. I've read that these cells are from cord tissue, but how many cord tissue have you analized? We know that there is a problem with the variability of behaviours of MSC, expecially cultured with FBS, have you used always the same flask at the same passage?

2. Western Blotting are overexpressed. Were the wells loaded with the same lysate? Please explain better the lane. The densitometric analysis was performed on these three lanes or with different WB?

3. Please show the success of the siRNA in MSC with WB.

4. Describe better the abstract by enphasizing the conclusion. 

Thank you for your answer and your work.

Author Response

  • REVIEWER’S COMMENTS & SUGGESTIONS
  • # Reviewer 2

The reviewer commented, “Kim et al in this study wanted to study the involvement of SOCS3 protein in the therapeutic efficacy of Mesenchymal Stem Cells against Acute Lung Injury. In my opinion this paper is interesting and well done but lacks of some point that could be explain better.”

2-1) The reviewer stated, “MSC's source. I've read that these cells are from cord tissue, but how many cord tissue have you analyzed? We know that there is a problem with the variability of behaviours of MSC, especially cultured with FBS, have you used always the same flask at the same passage?”

-> We appreciate the reviewer’s insightful comment and apologize for our mistake in describing the source of the MSC used in the study. In the original manuscript we submitted, we mistakenly wrote that Wharton jelly (WJ) was the source of the MSCs, but we have now corrected this in the revised manuscript. The correct source of the MSCs used in this study is umbilical cord blood (UCB). Our laboratory has MSCs derived from several sources, including WJ and UCB, but only UCB-derived MSCs were used in this study. We selected a lot of UCB-derived MSCs from a single donor at passage 5‒6 for use in both in vitro and in vivo experiments. During cell culture, we used the same product lot numbers of cell culture materials such as FBS and cell culture flask. We have incorporated this into Methods and Materials section in the revised manuscript. (line 294-296 and 299-300)

2-2) The reviewer stated, “Western Blotting are overexpressed. Were the wells loaded with the same lysate? Please explain better the lane. The densitometric analysis was performed on these three lanes or with different WB?”

-> To show a more detailed view of Western blotting results, we have enlarged the X-ray film image of Western blot and have included the unedited original raw images (with a blue background) for quantitative analysis in Supplementary figure S1. The same lysates were loaded on the wells for detection of SOCS1‒3 and GAPDH. Densitometric analysis was performed on these three lanes in non-treated MSCs and LPS-treated MSCs (n = 3 per each analyses), and the results were normalized to GAPDH. The densitometric analysis of SOCS1‒3 levels normalized to GAPDH are shown in Figure 2C. We have incorporated this into the figure legend of Figure 2C in the revised manuscript.

2-3) The reviewer stated, “Please show the success of the siRNA in MSC with WB.”

-> As the reviewer recommended, we have added the result of Western blot showing SOCS3 expression was effectively downregulated in MSCs treated with SOCS3 siRNA compared to MSCs with scrambled (control) siRNA, in Supplementary figure S2.

2-4) The reviewer stated, “Describe better the abstract by emphasizing the conclusion.”

-> We appreciate the reviewer's suggestion for improving the quality of the manuscript. We have inserted a concluding sentence at the end of the abstract: “Taken together, our findings suggest that SOCS3 is an important mediator for macrophage modulation in anti-inflammatory properties of MSCs.”

Round 2

Reviewer 1 Report

The authors have addressed most of the points and have discussed what they weren't able to perform as limitations.

Reviewer 2 Report

The authors have addressed most of the points and for this reason I accept the paper in the present form